# Thallium Contamination of Drinking Water: Health Implications in a Residential Cohort Study in Tuscany (Italy)

**DOI:** 10.3390/ijerph18084058

**Published:** 2021-04-12

**Authors:** Daniela Nuvolone, Davide Petri, Maria Cristina Aprea, Silvano Bertelloni, Fabio Voller, Ida Aragona

**Affiliations:** 1Unit of Epidemiology, Regional Health Agency of Tuscany, 50124 Firenze, Italy; davidepetri.bio@gmail.com (D.P.); fabio.voller@ars.toscana.it (F.V.); 2Department of Occupational Toxicology and Industrial Hygiene, Public Health Laboratory, 53100 Siena, Italy; cristina.aprea@uslsudest.toscana.it; 3Pediatric Division, Department of Obstetrics, Gynecology and Pediatrics, University Hospital, 56126 Pisa, Italy; s.bertelloni@ao-pisa.toscana.it; 4Department of Prevention, Health Agency of North-West Tuscany, 56124 Pisa, Italy; ida.aragona@uslnordovest.toscana.it

**Keywords:** contaminated sites, drinking water, mines sites, mortality, residential cohort study, thallium, water safety plan

## Abstract

In 2014–2015, concentrations of thallium above the recommended reference value (EPA: 2 µg/L) were measured in some parts of the drinking water distribution system in the municipality of Pietrasanta (Tuscany, Italy). An extensive campaign of water samples and human biomonitoring surveys were implemented to quantify the exposure of population. A residential cohort epidemiological study was carried out on the population of the municipality of Pietrasanta, aimed at comparing the health status of residents in the areas affected by thallium contamination with residents living in the rest of the municipality. Cohort included people residing in the municipality of Pietrasanta from 1 January 2000 to 31 December 2015. Residence addresses were georeferenced and each subject living in one of the three contaminated areas were defined as exposed. Mortality, hospital discharge data and adverse pregnancy outcomes were taken from administrative health databases. Cox proportional hazard models and logistic models were used to test the association between thallium exposure and health outcome. This study did not show any excess of risk in terms of mortality and hospitalization in the population residing in the areas served by thallium-contaminated aqueduct branches, compared to the rest of the not contaminated area. Increased risks for low birth weight (OR = 1.43 95% CI 0.91–2.25) and pre-term birth (OR = 1.40 95% CI 0.82–2.37) were observed. In view of the paucity of epidemiological studies on thallium, this study is an important contribution to the state of knowledge of the health effects of chronic exposures to low concentrations of thallium.

## 1. Introduction

Thallium (Tl) is a heavy metal that occurs widely at low concentrations in the earth’s crust [1,2]. In rare situations, due to both natural and anthropogenic sources, it may be found in the environment at higher concentrations. Cases of contamination of soils, water, crops have been reported [1,2,3]. Human activity has significantly increased its levels in the environment: it is used in the chemical industry as a catalyst and in synthesis processes, in the electrical and electronics industry, in the production of optical systems, in cement manufacture, and in medical procedures [1,2,3]. Thallium has historically been used as a rat poison and insecticide, as well as for suicide and murder, but due to frequent unintentional poisoning incidents, domestic use was banned from the mid-1970s in the USA and several other countries [2,4,5]. Human exposure occurs mainly by consuming contaminated food, drinking water, and other liquids [1,6,7,8]. Inhalation and dermal routes of exposure are less relevant. Cigarette smoking is considered a minor source of exposure [9,10]. Thallium enters the bloodstream rapidly and it is transported through the entire body, leading to accumulation in the bones, kidneys, and nervous system [11,12]. The acute effects of ingesting high doses of thallium are well known: gastric and intestinal ulcers, alopecia, and polyneuropathy are considered to be classic thallium poisoning syndromes [13,14,15,16]. Other symptoms include disorders such as insomnia, paralysis, loss of body mass, internal bleeding, myocardial injury and, consequently, death [13,14,15,16]. Cases of acute intoxication in humans are also reported in occupational epidemiology studies [17,18,19]. Peripheral sensory disturbances, mental disturbances, weight loss, and insomnia are most frequently observed in cases of acute intoxication during occupational accidents [17,18,19]. In the case of chronic intoxication, symptoms are similar to those of acute intoxication, but are generally milder, depending on the magnitude and duration of exposure [20,21,22,23,24]. There are insufficient data to assess mutagenic, carcinogenic, and teratogenic effects [1]. While the scientific literature provides ample evidence on the effects of acute and chronic intoxication with high doses of thallium, far fewer studies have addressed the effects of exposures to low to medium levels of thallium. The main objective of most studies has been to quantify thallium concentrations in human matrices, either to characterize the general population, not occupationally exposed, and thus have reference values [25,26,27,28,29,30,31,32], or to determine the impacts of exposure in areas where thallium is present, either due to proximity to industrial plants, mineral deposits, or due to contamination of water, soil, and crops [33,34,35,36,37,38,39].

In 2014–2015, concentrations of Tl above the recommended reference value (EPA: 2 µg/L) were measured in some parts of the drinking water distribution system in the municipality of Pietrasanta (Tuscany, Italy) [40,41]. Due to its rarity in nature, Tl is not included in the list of parameters for monitoring drinking water quality under European and Italian legislation. The source of contamination was immediately identified as a spring feeding the drinking water distribution network located in the immediate vicinity of an abandoned mining site. Although the contaminated source was quickly disconnected from the distribution network, contamination was detected due to the release of Tl from the contaminated pipelines. Therefore, the local authorities issued several ordinances prohibiting the use of water for drinking and food preparation in three areas of the municipality, first in the hamlet of Valdicastello Carducci, then the historic center, and lastly the hamlet of Pollino. The pipelines were completely or partially replaced, or flushed using experimental and innovative techniques. These emergency measures made it possible to bring thallium values in the distribution network below the recommended limits and to lift the ordinances.

In this context, in addition to a comprehensive water sampling campaign in the affected area and a large neighboring area, a series of human biomonitoring campaigns were also initiated to assess the impact of contamination on the exposure levels of the resident population [40]. A total of 2154 urine samples and 254 hair samples were analyzed in different areas affected by the contamination and at different times following the issuing/lifting of the ordinances. Moreover, individual information on lifestyle, occupation, risk factors, and symptomatology potentially related to thallium exposure was also collected. The sample selection criteria and the results of the biomonitoring campaigns are described elsewhere [40].

At the same time, a residential cohort epidemiological study was carried out on the population of the municipality of Pietrasanta, aimed at comparing the health status of residents in the areas affected by thallium contamination with that of those who lived in the rest of the municipality. The aim of this study is, therefore, to assess the medium- to long-term health effects of chronic exposure to low to medium levels of thallium, through the reconstruction of residential histories and data from administrative health database (mortality, hospitalizations, and adverse pregnancy outcomes).

## 2. Materials and Methods

### 2.1. Evaluation of Thallium Exposure

Thallium exposure was defined on the basis of the thallium values found in the numerous water samples from the aqueduct network carried out by the water supplier and the local health authority, about 4800 samples in the two years following the first episode of contamination. The highest levels of thallium in water were measured in the Valdicastello area, the area closest to the contaminated source: up to 79.5 µg/L. In the historic center of Pietrasanta and the hamlet of Pollino levels were, respectively, up to 6.80 µg/L, and 8.40 µg/L. The Valdicastello area shows an increasing geographical gradient as it nears the mining site located in the highest part of the hamlet (Figure 1): in fact, in the upper part, thallium concentrations of up to 79.5 µg/L were observed, in the central area up to 55.4 µg/L, and in the lower part up to 43.8 µg/L. The samples taken in the remaining areas of the municipality of Pietrasanta and in the surrounding areas showed thallium values below the recommended limits. Therefore, in this study the citizens living at residence addresses that were affected by the drinking water prohibition ordinances in the period October 2014 to July 2015 are defined as exposed. The map (Figure 1) shows the water utility users affected by the prohibition ordinances. In red those of the hamlet of Valdicastello, in yellow the area of the historic center of Pietrasanta and in green the Pollino area. Other water utility users in the municipality of Pietrasanta not affected by the drinking water contamination episodes are shown in grey.

### 2.2. The Definition of the Cohort

The study cohort consists of residents registered in the registry office of the municipality of Pietrasanta on 1 January 2000, and of all those subsequently registered as residents in the municipality, by birth or immigration, up to 31 March 2015. The reconstructed cohort is open and dynamic, since the following were taken into account: births and deaths of residents in the municipality, which occurred during the entire observation period; migratory movements out of and into the municipality; and changes in residence addresses within the municipality.

The archive of residents provided by the Pietrasanta registry office was subjected to quality control procedures, with the correction of cases that had been improperly duplicated and the exclusion of persons enrolled in the registry of Italians resident abroad. For each subject that joined the cohort, the residential history was reconstructed, i.e., all changes of residence up to 31 March 2015 were recorded; for subjects that emigrated from the municipality before this date, the observation period ended on the date of emigration.

Geographical coordinates were attributed to each residential address of the subjects in the study, managed through a geographic information system. The residential addresses were georeferenced using the geographical database of the Tuscany Region. The results of the georeferencing were subjected to data quality analysis, checking the degree of completeness, and the level of accuracy.

Each subject in the study cohort was also assigned the socioeconomic deprivation index (SES) value of the census section where their residence is located. This summary index takes into account the share of the population with education equal to or less than primary school leaving certificate, the active population who are unemployed or looking for their first job, the number of rented occupied dwellings, single-parent families with dependent children living together, and the housing density. The deprivation index was classified in three groups (high, medium, low), according to the tertiles of the distribution, with class “high” referring to high socioeconomic deprivation.

### 2.3. Follow-Up Procedures and Health Data

For each subject present in the cohort of residents in Pietrasanta on 1 January 2000, a follow-up of the survival status up to 15 March 2015 was carried out, using the municipal registry archives. For subjects who died from 2004 to 2012, the cause of death was attributed using data from the Tuscan Regional Mortality Register (RMR), which records the deaths of residents in the Tuscany region wherever they occurred, within and outside the region. The procedure for identifying deaths and the cause of death is carried out anonymously by means of a code in use by the Tuscany region, assigned to protect citizens’ privacy. The analysis of hospitalizations relative to the cohort of residents was carried out with data from the Hospital Discharge Forms (SDO) of the Tuscany region, relative to ordinary and day-hospital discharge, occurring within and outside the region in the period 2000–2014. A subject who was hospitalized several times for different causes was counted once for each hospitalization condition. In order to consider only incident cases, a person who was hospitalized several times for the same disease was counted only once, considering the first admission. Only the primary diagnosis at discharge was considered in the analysis. In order to assess certain adverse birth outcomes, in particular preterm birth (live births within 37 weeks’ gestation) and low birth weight (birth weight < 2500 g), the cohort subjects were also assigned information from birth attendance certificates covering the period 2001–2014. Unlike the mortality and hospitalization archives, which also record cases occurring outside the region, the birth certificates only record information on births occurring in Tuscany.

Causes of death and hospitalization diagnoses are classified according to the ninth revision of the International Classification of Diseases (ICD-9).

### 2.4. Statistical Analysis

Each resident of the cohort contributed to the calculation of the person-years at risk from 1 January 2000, if present in the municipality on this date, or from the date of his/her registration in the municipal registry if he/she entered the municipality after 1 January 2000, until the date of death/hospitalization, emigration, or end of follow-up. Descriptive statistics of the cohort at baseline were produced with respect to gender, age, socioeconomic deprivation index, and thallium exposure. The association between thallium exposure and mortality/morbidity was assessed with a Cox proportional hazard model, estimating relative hazards (Hazard Ratio—HR) and 95% confidence intervals (95% CI). The hazard proportionality assumption was tested for all categorical variables; if this assumption was violated, the effect estimates were compared using stratified Cox models for the covariate under study. As regards thallium exposure, two different models were conducted: in the first model, those exposed are defined as those who resided in the hamlet of Valdicastello, which is the area exposed to higher thallium values in the drinking water; in the second model, those exposed are defined as those who resided in one of the three areas of the municipality affected by the water distribution system contamination (Valdicastello, historic center, and Pollino), referred to as “overall exposure”. In the multivariate Cox models, the time axis is defined by age, and the confounding effect of gender, socioeconomic index, and calendar period was taken into account. The analysis also took into account the duration of residence and the models were restricted to long-term residents, i.e., those who have resided in the study area for at least 5 years.

The association between thallium exposure and adverse pregnancy outcomes was assessed using a multivariate logistic model, estimating Odds Ratio (OR) and 95% confidence intervals (95% CI). The multivariate logistic model takes into account the confounding effect of other risk factors for pre-term birth and low birth weight indicated on the birth certificates, such as the mother’s smoking habits during pregnancy, the mother’s educational qualification, the use of assisted reproduction, and multiple births.

## 3. Results

### 3.1. Characteristics of the Study Cohort

The cohort of residents under study consisted of 33,708 people, 47.3% men, for a total of 883,655 person-years. The main socio-demographic characteristics of the cohort members are shown in Table 1.

Fifty-two percent of residents had a high socioeconomic deprivation index, 68% of subjects already lived in the municipality of Pietrasanta on 1 January 2000, and 47% were born in the municipality of Pietrasanta. The percentage of residents born abroad was 9%. At the end of the follow-up, 69% of the subjects were alive and resident, 19% had emigrated, and 3442 subjects (10%) had died.

Table 2 shows the main socio-demographic characteristics of the residents in the two exposure groups, i.e., residents in the hamlet of Valdicastello in model 1 and residents in the overall exposure area in model 2. (Valdicastello, historic center, and Pollino). No significant differences were observed for gender and average age. Conversely, socioeconomic deprivation index was significantly associated with thallium exposure (*p* < 0.0001 in both models), although with opposite patterns when considering residents in the Valdicastello area alone (model 1) or residents in one of the three areas affected by the contamination (model 2). In the Valdicastello area the proportion of residents with a high socioeconomic deprivation index was significantly lower than observed in the rest of the municipality, while in the overall exposure area the proportion of people with a high deprivation index was significantly higher than the reference. This situation was closely correlated with the greater presence of a population of foreign origin (*p* < 0.0001) in the overall exposure area.

### 3.2. Association between Residence in Thallium Exposure Areas and Mortality

In the period 2000–2015, a total of 3442 deaths were recorded in the municipality of Pietrasanta, of which 645 (19%) occurred in one of the three exposure areas, and 138 (4%) in the hamlet of Valdicastello. There were 2638 deaths (77% of the total) between 2004 and 2012, the period for which it was possible to recover the cause of death. Of these, 492 occurred in one of the three thallium exposure areas and 97 in the hamlet of Valdicastello.

Table 3 shows the HR for all-cause and cause-specific mortality and the 95% confidence intervals for the two separate models. In both models the reference is the rest of the municipality of Pietrasanta. Overall mortality was significantly lower in exposed subjects than the reference, in both multivariate models (model 1: HR = 0.76, 95% CI: 0.64–0.91; model 2: HR = 0.88, 95% CI: 0.81–0.96). Significantly negative associations were also observed in both models for mortality from all cancers (model 1: HR = 0.60, 95% CI: 0.42–0.86; model 2: HR = 0.81, 95% CI: 0.68–0.96), particularly for lung cancer (model 1: HR = 0.23, 95% CI: 0.06–0.92; model 2: HR = 0.58, 95% CI: 0.36–0.92), and in the overall exposure models for diabetes (HR = 0.34, 95% CI: 0.16–0.74) and for diseases of the digestive system (HR = 0.60, 95% CI: 0.37–0.99). Although at non-significant levels, in the multivariate models with overall exposure an excess risk of mortality was shown for bladder (HR = 2.07, 95% CI: 0.92–4.63) and liver cancer (HR = 1.21, 95% CI: 0.79–1.87). The lower numbers in the models that include the exposure of residents of the hamlet of Valdicastello resulted in more unstable and inaccurate estimates.

The gender-stratified analysis reduced the statistical power and did not show any particular additional results, except that the excess mortality from bladder cancer was concentrated in the female population. The analysis limited to long-term residents, i.e., those who had lived there for at least five years, also confirmed the results from general population.

### 3.3. Association between Residence in Exposure Areas and Hospitalizations

Table 4 shows the first hospitalization risks (HR) and 95% confidence intervals of residents in the thallium exposure areas compared to those in the rest of the Pietrasanta municipality. In line with the mortality analysis, significant risk reductions associated with residence in contaminated areas were observed for several diseases. Negative associations are reported in both models for overall cancer hospitalizations (model 1: HR = 0.70, 95% CI: 0.57–0.86; model 2: HR = 0.85, 95% CI: 0.77–0.95), for diseases of the nervous system (model 1: HR = 0.74, 95% CI:0.60–0.92; model 2: HR = 0.88, 95% CI: 0.78–0.98), circulatory system (model 1: HR = 0.75, 95% CI: 0.65–0.87; model 2: HR = 0.85, 95% CI: 0.79–0.92), and genitourinary system (model 1: HR = 0.79, 95% CI: 0.67–0.94; model 2: HR = 0.90, 95% CI: 0.82–0.99), and only in models with overall exposure for respiratory diseases (model 2: HR = 0.90, 95% CI: 0.82–0.98), for stomach cancer (model 2: HR = 0.47, 95% CI: 0.24–0.94), and for congenital malformations (model 2: HR = 0.72, 95% CI: 0.55–0.93). Analyses stratified by gender and duration of residence confirm these results.

### 3.4. Association between Residence in Exposure Areas and Adverse Pregnancy Outcomes

There were 2629 births in the municipality of Pietrasanta in the period 2000–2015. Of these, 2102 can also be found in the archive of birth attendance certificates, which have been officially available since 2001 and were updated in 2014, when the study was conducted. Of the 2102 births, 1056 were male and 1046 female, 117 were resident in the hamlet of Valdicastello, and 393 in one of the three exposure areas.

The prevalence of low weight births was 5.2% in the hamlet of Valdicastello, 7.8% in the overall exposure area, and 6.1% in the rest of the municipality of Pietrasanta. The prevalence of pre-term births was 2.6% in the hamlet of Valdicastello, 5.7% in the overall exposure area and 4.5% in the rest of the municipality.

The results of the multivariate logistic analysis are shown in Table 5.

In the areas affected by thallium contamination there was an increased risk of 43% for low birth weight (model 2: OR = 1.43, 95% CI 0.91–2.25) and 40% (model 2: OR = 1.40, 95% CI 0.82–2.37) for pre-term birth. For the hamlet of Valdicastello, the number of cases of low birth weight and pre-term births is so small that it was not possible to reliably interpret the results.

## 4. Discussion

The aim of this study was to assess the health status of residents in the areas of the Pietrasanta municipality affected by the thallium contamination of drinking water. This event was a rare and exceptional occurrence that required emergency measures to bring thallium levels in drinking water down to safe levels and to stop public exposure. While the isolation of the contaminated spring from the distribution network, the flushing and total replacement of sections of piping in the aqueduct network caused an inconvenience to the communities concerned, it resulted in lower thallium levels and meant that ordinances prohibiting the use of drinking water for food purposes could be lifted. The thallium incident in the municipality of Pietrasanta was all the more exceptional considering how rare this element is, so much so that it is not yet included in the parameters for drinking water according to the regulations. 

### 4.1. Mortality and Morbidity Outcomes

The residential cohort study conducted in the municipality of Pietrasanta did not show any particular excess risk in terms of mortality and hospitalization in the population residing in the areas served by thallium-contaminated aqueduct branches, compared to the rest of the municipality of Pietrasanta. On the contrary, there was more evidence of more favorable epidemiological indicators in the group of residents in contaminated areas. The low number of residents in the hamlet of Valdicastello, i.e., the hamlet where the highest values of thallium in drinking water were recorded, considerably reduced the statistical power, although analyses of the major groups of pathologies (all-cause mortality, mortality and hospitalizations for cancer, and cardiovascular diseases) seem to confirm this trend. These results seem to be consistent with the results of the human biomonitoring campaigns carried out in the study area [40]. Urinary thallium values had a geometric mean of 0.467 µg/L (95th percentile equal to 1.89 µg/L) for the most exposed group (samples of residents in the hamlet of Valdicastello within two weeks of the drinking water prohibition ordinance), compared to 0.186 µg/L (95th percentile 0.563 µg/L) in the control group. The urinary thallium values found in the most exposed population groups in the municipality of Pietrasanta are slightly higher than the reference values defined by the various national and international societies and bodies. The most up-to-date assessments of reference values for metals in human matrices for the non-occupationally exposed Italian population are produced by the Società Italiana Valori di Riferimento (Italian Society for Reference Values—SIVR). In the latest 2017 edition, a geometric mean of 0.203 μg/L and a 95th percentile of 0.759 μg/L are reported as urinary reference values for thallium [25]. In the Pietrasanta study, 30.5% of the samples in the most exposed group had a value above the 95th percentile of the SIVR. Comparison with other international biomonitoring campaigns in US, Canada, and Europe show similar results [9,30,31,32,42,43,44].

Findings from Pietrasanta study can also be compared with a few studies conducted in areas with specific sources of thallium exposure, despite they have different study design. The first, in chronological order, is the German study by Brockhaus et al. [34] in 1981 on a population living near a cement factory with known thallium emissions. The mean urinary thallium level in the German study population was higher than value observed in Valdicastello, 2.6 μg/L and 0.467 μg/L, respectively. In the German study authors did not observe any associations with self-reported symptoms and diseases, except for sleep disturbance, tiredness, headache and other psychic alterations. A very different situation was described in the Chinese study by Xiao et al. [35] in a rural area of China with natural thallium contamination of the soil. In this case, the local population showed clear symptoms of thallium poisoning in the period 1960–1970, such as fatigue, muscular pain, visual disturbances, and hair loss, with a total of 189 cases of poisoning [35]. Early investigations showed thallium values in drinking water to be quite high, in the range 3.7–40 μg/L [35]. Above all, high concentrations of thallium were determined in cabbages produced in local gardens in soils with very low pH values. The results of the biomonitoring campaigns were alarming: in the three contaminated villages, the range of urinary thallium values was 2.5–2668 μg/L, with a mean of 521.9 μg/L [36]. A study by Croatian researchers investigated the presence of thallium and uranium in an eastern part of Croatia, which is affected by the presence of high concentrations of various metals in drinking water, particularly due to the population’ use of unmonitored private wells [39]. The values obtained were very low, with 0.15 μg/L as the highest geometric mean of urinary levels recorded in Draž [39].

With the exception of the Croatian study, the urinary thallium values observed in the Pietrasanta samples were thus significantly lower than the few international experiences available on studies in areas affected by specific sources of thallium contamination. Moreover, albeit with due caution due to the paucity of available studies, especially on the effects of chronic low-dose exposures, the WHO has also provided estimates of thallium levels that are potentially harmful to human health. In the 1996 document of the International Programme on Chemical Safety “Environmental Health Criteria 182. Thallium” [1] the working group of international experts, commissioned to conduct a literature review of all available thallium studies, both experimental laboratory and human observational epidemiological studies, concludes that exposure resulting in a urinary thallium concentration below 5 µg/L is not likely to cause adverse health effects. In the range 5–500 µg/L the magnitude of risk and severity of adverse effects are uncertain, while exposure leading to urinary concentrations above 500 µg/L is associated with poisoning. The mean urinary thallium levels observed in the Pietrasanta area were well below the value of 5 µg/ L. 

### 4.2. Adverse Pregnancy Outcomes

The only item of note from Pietrasanta study, although affected by low statistical power, is the increased risk of low birth weight and pre-term birth associated with residence in the exposure areas. This result, although weak from a statistical point of view, deserves particular interest in the light of the results of some studies conducted in China. In particular, a 2016 Chinese study looked at the relationship between levels of certain metals, including thallium, in the mother’s urine and low birth weight [37]. This was a case-control study, following a pilot study by Hu et al. [38], involving 816 pregnant women (including 204 cases of low birth weight and 612 controls) living in Hubei province. The median urinary values of thallium in women with a low birth weight delivery were 0.64 μg/g creatinine (range: <LOD (Limit of Detection)-8.15 μg/g creatinine), while median values of 0.55 μg/g creatinine (range: <LOD-6.90 μg/g creatinine) were observed in controls. These values are similar to those found in the Pietrasanta study. In the multivariate logistic regression model, increasing levels of thallium (distribution by tertiles) resulted in a 90% increased risk of low weight (OR = 1.90 95% CI 1.01–3.58), taking into account gestational age, income, maternal body mass index, passive smoking, parity, and hypertension during pregnancy. Therefore, although evidence of association is limited to a few studies, the issue of the effects of thallium, even at low to medium doses, on adverse pregnancy outcomes undoubtedly merits further investigation, considering foetuses may be more susceptible to environmental noxae than adults and exposures in utero can increase the risk of developmental delays, adult chronic illnesses, and next generation effects [45,46,47].

### 4.3. Strenghts and Limitations of the Study

To date, the Pietrasanta study is the only study available internationally on the long-term effects of exposure to low to medium concentrations of thallium, assessed using a longitudinal approach. In addition to the cohort design that allows an individual-level assessment of time at risk, the strengths of this study include the accurate reconstruction of a dynamic, open cohort of over 30,000 people. Each resident was followed over a long period of time and all demographic movements and changes of residence addresses were considered. The geo-referencing system, based on a complete and accurate geographical database, provided high-quality and complete geocoded addresses, minimizing the risk of exposure misclassification. The use of reliable health datasets, by linking them with mortality, hospitalization, and birth certificate records, also ensured high quality health data. The causes of mortality/morbidity chosen were quite general and included a broad spectrum of diseases due to the lack of specific literature on the effects of chronic exposure to low levels of thallium.

As this was a population-wide residential study, no data were available on the personal habits of the people included in the cohort, particularly those risk factors that have a known etiological role for many of the diseases investigated, such as cigarette smoking, alcohol, physical activity, and obesity. To at least partly remedy this shortcoming, socioeconomic deprivation index was used as a confounding factor in the multivariate models. This is because socioeconomic index, as well as being in itself a determinant of health status, is also a proxy for individual risk behaviors. This is a commonly used approach in environmental epidemiology, and particularly in residential cohort studies. Nevertheless, having used an aggregated socioeconomic indicator at census section level does not solve the problem of the ecological fallacy that may have introduced a misclassification at individual level. The most critical element of this study, however, is the lack of historical thallium exposure data. The fact that thallium was not included in the parameters for assessing potability under the regulations prevented an accurate reconstruction of thallium exposure. However, the hypothesis of long-term exposure seems to be supported by some evaluations. The Local Health Authority Public Health Laboratory’s retrieval of data from historical dialysis water archives confirmed the presence of thallium values > 2 μg/L even in samples taken in 2011. However, above all, the level of contamination of the pipes, which has in fact necessitated the total replacement of the distribution network where possible, or alternatively the use of innovative cleaning techniques, supports the hypothesis of chronic exposure of the resident population. Biomonitoring data from hair samples, which reflect a longer duration of exposure than urine samples, also show higher levels in the three exposure areas than observed in the control group [40,41].

The case of thallium contamination of the drinking water distribution network in some areas of the municipality of Pietrasanta has raised a number of critical issues. In a context of strong public alarm, the emergency was tackled by means of close synergy and multidisciplinary cooperation between the various players involved: citizens, administrators, health authorities, managing body, technicians, and researchers. It also highlighted the limitations of the old approach to managing and protecting drinking water, which was definitively replaced by the new Commission Directive 2015/1787/EC, which promotes a new holistic approach that shifts the focus from retrospective monitoring of distributed water to risk prevention and management in the drinking water supply chain, along the lines of the Water Safety Plans (WSPs) drawn up by the World Health Organization as early as 2004. The WSP model pursues integrated risk assessment and management, extending from capture to tap, in order to protect the source water resources and control the system and processes to ensure the long-term absence of potential physical, biological, and chemical hazards.

## 5. Conclusions

In conclusion, this study conducted in the municipality of Pietrasanta highlights that the level of population thallium exposure, due to the contamination of the drinking water system, did not resulted associated with noticeable epidemiological effects, in terms of mortality and morbidity. The associations observed with adverse pregnancy outcomes merits further investigations. In view of the paucity of epidemiological studies on thallium, this study makes an important contribution to the state of knowledge of the health effects of chronic exposures to low concentrations of thallium.

## Figures and Tables

**Figure 1 ijerph-18-04058-f001:**
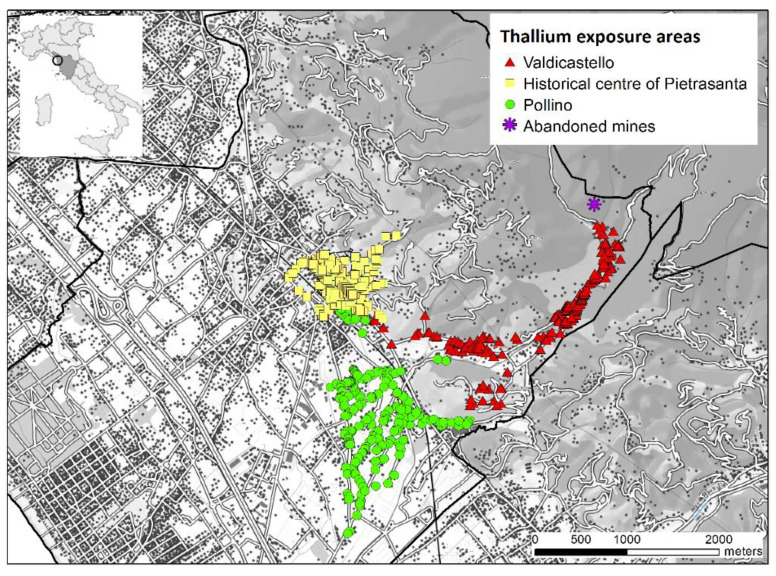
Map of the study area.

**Table 1 ijerph-18-04058-t001:** Main characteristics of the cohort of residents in the municipality of Pietrasanta.

		MEN		WOMEN		TOTAL	
		N.	%	N.	%	N.	%
	Total	15,940	100	17,768	100	33,708	100
Age Group (Years) at Baseline							
<1		1377	8.6	1380	7.8	2757	8.2
1–9		1263	7.9	1256	7.1	2519	7.5
10–19		1244	7.8	1233	6.9	2477	7.3
20–29		2104	13.2	2289	12.9	4393	13.0
30–39		2915	18.3	2980	16.8	5895	17.5
40–49		2328	14.6	2342	13.2	4670	13.9
50–59		1886	11.8	2134	12.0	4020	11.9
60–69		1618	10.2	1997	11.2	3615	10.7
70–79		966	6.1	1523	8.6	2489	7.4
≥80		239	1.5	634	3.6	873	2,6
Socioeconomic Deprivation Index (SES)							
Low		2595	16.3	3053	17.2	5648	16.8
Medium		4910	30.8	5507	31.0	10,417	30.9
High		8431	52.9	9205	51.8	17,636	52.3
Missing		3	0.02	4	0.02	7	0.02
Calendar Period							
Present at 1 January 2000		10,894	68.3	12037	67.7	22,931	68.0
Entered between 2000 and 2004		1461	9.2	1594	9.0	3055	9.1
Entered between 2005 and 2009		1890	11.9	2162	12.2	4052	12.0
Entered between 2010 and 2015		1695	10.6	1975	11.1	3670	10.9
Birth Place							
Pietrasanta		7628	47.9	8096	45.6	15,724	46.6
Tuscany		5579	35.0	6072	34.2	11,651	34.6
Italy		1568	9.8	1844	10.4	3412	10.1
Foreign		1165	7.3	1756	9.9	2921	8.7
Vital Status at 15 March 2015							
Alive		10,837	68.0	12,297	69.2	23,134	68.6
Emigrant		3059	19.2	3399	19.1	6458	19.2
Dead		1669	10.5	1773	10.0	3442	10.2
Other		375	2.4	299	1.7	674	2.0

**Table 2 ijerph-18-04058-t002:** Main characteristics of the cohort by residence in the three Tl exposure areas.

	Model 1: Residence in Valdicastello		Model 2: Overall Exposure		Not Contaminated Area	
	N.	%	N.	%	N.	%
	1682	100	6854	100	26,854	100
Gender						
Men	772	45.9	3232	47.1	12,708	47.3
Women	910	54.1	3622	52.9	14,146	52.7
Age at Baseline						
Mean (SD)	48.5 (23.4)		49.7 (23.5)		50.9 (23.9)	
Socioeconomic Deprivation Index (SES)						
Low	654	38.9	705	10.9	4897	18.2
Medium	704	41.9	2110	30.8	8308	30.9
High	324	19.3	3987	58.2	13,649	50.8
Calendar Period						
Present at 1 January 2000	1135	67.5	4288	62.6	18,643	69.4
Entered between 2000 and 2004	169	10.0	801	11.7	2254	8.4
Entered between 2005 and 2009	187	11.1	912	13.3	3140	11.7
Entered between 2010 and 2015	191	11.4	853	12.4	2817	10.5
Birth Place						
Pietrasanta	954	56.7	3580	52.2	12,144	45.2
Tuscany	467	27.8	1878	27.4	9773	36.4
Italy	118	7.0	667	9.7	2745	10.2
Foreign	143	8.5	729	10.6	2192	8.2

**Table 3 ijerph-18-04058-t003:** Association between residence in the thallium exposure area and all-cause and cause-specific mortality.

		Model 1: Residence in Valdicastello				Model 2: Overall Exposure *			
Cause of Death (ICD-9)	Reference N.	N.	HR **	95% CI		N.	HR	95% CI	
All Causes (001–899)	2797	138	0.76	0.64	0.91	645	0.88	0.81	0.96
Malignant Neoplasms (140–209)	764	31	0.60	0.42	0.86	157	0.81	0.68	0.96
Stomach (151)	47	2	0.67	0.16	2.77	9	0.77	0.38	1.58
Colon Rectum (153,154,159)	97	4	0.62	0.23	1.69	21	0.85	0.53	1.37
Liver (155,156)	89	6	0.92	0.40	2.11	27	1.21	0.79	1.87
Pancreas (157)	49	1	0.31	0.04	2.25	8	0.63	0.30	1.34
Lung (162)	141	2	0.23	0.06	0.92	21	0.58	0.36	0.92
Breast (174,175)	43	1	0.29	0.04	2.11	11	0.99	0.51	1.93
Ovary (183)	18	1	0.70	0.09	5.26	4	0.89	0.30	2.65
Bladder (188)	18	3	1.81	0.53	6.16	9	2.07	0.92	4.63
Lymphatic and Hematopoietic Tissue (200–208)	66	4	0.96	0.35	2.66	19	1.09	0.65	1.82
Diabetes Mellitus (250)	75	1	0.25	0.03	1.78	7	0.34	0.16	0.74
Nervous System and Sense Organs (320–289)	96	3	0.53	0.17	1.67	20	0.79	0.49	1.29
Circulatory System (390–459)	836	48	0.89	0.66	1.19	217	0.97	0.84	1.13
Ischemic Heart Disease (410–414)	228	19	1.18	0.73	1.89	66	1.13	0.86	1.48
Respiratory System (460–519)	137	8	0.96	0.47	1.97	38	1.03	0.72	1.48
COPD (490–492,494–496)	71	3	0.73	0.23	2.35	19	1.02	0.61	1.69
Digestive System (520–579)	113	3	0.43	0.14	1.37	18	0.60	0.37	0.99
Chronic Liver Disease and Cirrhosis (571)	46	0	n.d.	n.d.	n.d.	6	0.49	0.21	1.15
Genitourinary System (580–629)	35	2	0.94	0.22	4,00	11	1.15	0.59	2.28
Renal Failure (584–586)	21	0	n.d.	n.d.	n.d.	4	0.71	0.24	2.08

* Overall exposure: residence in one of the three contaminated areas: Valdicastello, historical center and Pollino. ** HR: Hazard Ratio from Cox proportional hazard models stratified by follow-up periods (2000–2004, 2005–2009, and 2010–2015) and adjusted by gender, age, and socioeconomic deprivation index. Reference: not contaminated areas in the municipality of Pietrasanta. Years 2000–2015 for all-cause mortality and 2004–2012 for cause-specific mortality.

**Table 4 ijerph-18-04058-t004:** Association between residence in the thallium exposure area and cause-specific hospitalization.

		Model 1: Residence in Valdicastello				Model 2: Overall Exposure *			
Diagnoses (ICD-9)	Reference N.	N.	HR **	95% CI		N.	HR **	95% CI	
Malignant Neoplasms (140–209)	1926	96	0.70	0.57	0.86	425	0.85	0.77	0.95
Stomach (151)	75	1	0.21	0.03	1.51	9	0.47	0.24	0.94
Colon Rectum (153,154,159)	229	13	0.78	0.44	1.36	58	1.00	0.75	1.34
Liver (155,156)	123	8	0.93	0.45	1.90	28	0.91	0.60	1.37
Pancreas (157)	43	2	0.67	0.16	2.76	9	0.83	0.40	1.70
Larynx (161)	51	3	0.94	0.29	3.04	15	1.19	0.67	2.12
Lung (162)	165	8	0.76	0.37	1.55	36	0.85	0.59	1.23
Connective and Other Soft Tissue (171)	8	1	1.58	0.20	12.75	4	1.96	0.59	6.53
Breast (174,175)	270	14	0.63	0.37	1.07	69	0.97	0.75	1.27
Ovary (183)	31	3	1.13	0.35	3.71	9	1.10	0.52	2.33
Bladder (188)	187	13	1.08	0.61	1.90	40	0.87	0.62	1.22
Kidney (189)	73	5	1,00	0.40	2.48	13	0.75	0.41	1.35
Central Nervous System (191,192, 225, 2396)	81	7	1.24	0.57	2.71	14	0.68	0.39	1.20
Thyroid (193)	70	6	1.28	0.55	2.96	13	0.73	0.40	1.32
Lymphatic and Hematopoietic Tissue (200–208)	168	10	0.87	0.46	1.65	39	0.90	0.63	1.27
Diabetes Mellitus (250)	136	8	0.89	0.43	1.82	21	0.60	0.38	0.94
Nervous System and Sense Organs (320–289)	1632	87	0.74	0.60	0.92	374	0.88	0.78	0.98
Circulatory System (390–459)	3636	192	0.75	0.65	0.87	798	0.85	0.79	0.92
Ischemic Heart Disease (410–414)	1070	56	0.77	0.59	1.01	252	0.92	0.80	1.06
Respiratory System (460–519)	2550	163	0.88	0.75	1.03	590	0.90	0.82	0.98
COPD (490–492,494–496)	882	64	0.99	0.77	1.28	200	0.87	0.75	1.02
Acute Respiratory Infections (460–466)	135	6	0.72	0.32	1.63	27	0.75	0.50	1.14
Digestive System (520–579)	3239	197	0.85	0.73	0.98	772	0.93	0.86	1.00
Chronic Liver Disease and Cirrhosis (571)	238	13	0.74	0.42	1.29	66	1.08	0.82	1.42
Genitourinary System (580–629)	2364	143	0.79	0.67	0.94	554	0.90	0.82	0.99
Renal Failure (584–586)	288	11	0.57	0.31	1.04	59	0.79	0.60	1.05
Congenital Anomalies (740–759)	374	20	0.74	0.47	1.16	67	0.72	0.55	0.93

* Overall exposure: residence in one of the three contaminated areas: Valdicastello, historical center and Pollino. ** HR: Hazard Ratio from Cox proportional hazard models stratified by follow-up periods (2000–2004, 2005–2009, 2010–2014) and adjusted by gender, age and socioeconomic index. Reference: not contaminated areas in the municipality of Pietrasanta. Years 2000–2014.

**Table 5 ijerph-18-04058-t005:** Association between residence in thallium exposure areas and adverse pregnancy outcomes.

.	Reference	Model 1: Residence in Valdicastello				Model 2: Overall Exposure *			
	N.	N.	OR **	95% CI		N.	OR	95% CI	
Low birth weight	102	6	0.90	0.38	2.12	30	1.43	0.91	2.25
Preterm birth	75	3	0.55	0.17	1.78	22	1.40	0.82	2.37

* Overall exposure: residence in one of the three contaminated areas: Valdicastello, historical center and Pollino. ** OR: Odds ratio from multivariate logistic regression models, adjusted by smoking habits of the mother, mother’s education and multiple births. Reference: not contaminated areas in the municipality of Pietrasanta. Years 2001–2014.

## Data Availability

There are legal restrictions on sharing a de-identified data set because data contain potentially identifying and sensitive people information (age, gender, residence, and health status).

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
