# Peer review of "Thallium Contamination of Drinking Water: Health Implications in a Residential Cohort Study in Tuscany (Italy)"

_ijerph, 2021, doi:10.3390/ijerph18084058_

Round 1
Reviewer 1 Report
The manuscript aimed to assess the medium- to long-term health effects of chronic exposure to low to medium levels of thallium, through the reconstruction of residential histories and data from administrative health database. The sampling startegy as well methods used for data analysis were well detailed in a step wise way. However the manuscripts can still be improved by addressing these comments:
. The keywords must be arranged alphabetically
. In the discussion section the relevant literature must be used to support related results from the study, however some parts of the discussion reads more like a literature review than discussion section.
.The conclusion could be better improved by highlighting the main findings in the discussion rather the conclusion being written in a generic manner.
Reviewer 2 Report
Line 13: Change the word “activated”
L15” with that of those” rephrase it
L36: add its before levels
L36-38: It needs to be rephrased
L46” bones, kidneys and nervous system.” Any reference? Example?
L194: “15940 men and 17768” show by percentage
Table 1: show the population number as % not by numbers, what the Person-Years means? I recommend showing the 0 years old by less than 1
The person in years does not make sense to me. DO you mean 18900 kids enter to the municipality? If entered what's the relation with thallium
What d you mean by” Socio-economic status “ It is not clear
Table 3 is not clear. Authors must clean the table with two decimal and removing irrelevant disease or infrequent cases. This applies to all other tables as well.
Reviewer 3 Report
This is a high quality manuscript. However, the section "Discussion" can be divided into several subsections.
